# Sustained Systemic Levels of IL-6 Impinge Early Muscle Growth and Induce Muscle Atrophy and Wasting in Adulthood

**DOI:** 10.3390/cells10071816

**Published:** 2021-07-18

**Authors:** Laura Pelosi, Maria Grazia Berardinelli, Laura Forcina, Francesca Ascenzi, Emanuele Rizzuto, Marco Sandri, Fabrizio De Benedetti, Bianca Maria Scicchitano, Antonio Musarò

**Affiliations:** 1DAHFMO-Unit of Histology and Medical Embryology, Sapienza University of Rome, Via A. Scarpa, 14, 00161 Rome, Italy; laura.pelosi@uniroma1.it (L.P.); mariagrazia.berardinelli@gmail.com (M.G.B.); laura.forcina@uniroma1.it (L.F.); 2Department of Clinical and Molecular Medicine, Risk Management Q and A, Sant’Andrea Hospital, “Sapienza” University, 00161 Rome, Italy; francesca.ascenzi@uniroma1.it; 3Department of Mechanical and Aerospace Engineering, Sapienza University of Rome, 00184 Rome, Italy; emanuele.rizzuto@uniroma1.it; 4Veneto Institute of Molecular Medicine, 35129 Padua, Italy; marco.sandri@unipd.it; 5Department of Biomedical Sciences, University of Padova, 35121 Padua, Italy; 6Division of Rheumatology and Immuno-Rheumatology Research Laboratories, Bambino Gesù Children’s Hospital, 00146 Rome, Italy; fabrizio.debenedetti@opbg.net; 7Istituto di Istologia ed Embriologia, Università Cattolica del Sacro Cuore, Fondazione Policlinico Universitario “Agostino Gemelli”, IRCCS, 00168 Rome, Italy; biancamaria.scicchitano@unicatt.it; 8Laboratory Affiliated to Istituto Pasteur Italia—Fondazione Cenci Bolognetti, DAHFMO-Unit of Histology and Medical Embryology, Sapienza University of Rome, Via Antonio Scarpa, 14, 00161 Rome, Italy; 9Scuola Superiore di Studi Avanzati Sapienza (SSAS), Sapienza University of Rome, 00185 Rome, Italy

**Keywords:** interleukin-6, skeletal muscle, muscle growth, muscle atrophy, PGC-1α

## Abstract

IL-6 is a pleiotropic cytokine that can exert different and opposite effects. The muscle-induced and transient expression of IL-6 can act in an autocrine or paracrine manner, stimulating anabolic pathways associated with muscle growth, myogenesis, and with regulation of energy metabolism. In contrast, under pathologic conditions, including muscular dystrophy, cancer associated cachexia, aging, chronic inflammatory diseases, and other pathologies, the plasma levels of IL-6 significantly increase, promoting muscle wasting. Nevertheless, the specific physio-pathological role exerted by IL-6 in the maintenance of differentiated phenotype remains to be addressed. The purpose of this study was to define the role of increased plasma levels of IL-6 on muscle homeostasis and the mechanisms contributing to muscle loss. Here, we reported that increased plasma levels of IL-6 promote alteration in muscle growth at early stage of postnatal life and induce muscle wasting by triggering a shift of the slow-twitch fibers toward a more sensitive fast fiber phenotype. These findings unveil a role for IL-6 as a potential biomarker of stunted growth and skeletal muscle wasting.

## 1. Introduction

Interleukin-6 (IL-6) is a soluble mediator with pleiotropic actions on different tissues and organs. The multifaceted nature of IL-6 action can be ascribed to the multi-level regulation of the different signaling pathways that it is able to induce. Although IL-6 is produced by several cell types and it can signal through different receptor complexes, inducing a variety of intracellular pathways, its expression and signal-transduction is strictly controlled. IL-6 expression is regulated at the transcriptional and post-transcriptional level, while the induction of the signaling cascade is negatively regulated by systems of internalization/degradation of receptor subunits and by the action of negative feedback molecules [1,2]. Thus, acting as a warning molecule, the abundance and persistence of IL-6 need a fine regulation in order to limit the powerful action of the cytokine at the physiologic stimuli. For instance, in skeletal muscle, the local and transient production of IL-6 in response to physical exercise plays an important role in regulating homeostatic processes such as satellite cell proliferation and myonuclear accretion [3,4]. Indeed, it has been extensively described that under physiological stimuli, such as exercise, IL-6 levels remained elevated for several hours to several days acting both locally, in an autocrine or paracrine manner leading to increase glucose uptake and fat oxidation, or in a hormone-like fashion, increasing hepatic glucose production and lipolysis in adipose tissue [5,6,7,8,9,10]. Upon stimulus termination, circulating IL-6 levels return to pre-stimulus conditions, which are normally very low or undetectable. On the other hand, deregulated amounts of systemic IL-6 have been observed in cases of persistent distresses. In particular, the chronic persistence of systemic IL-6 is associated with several pathophysiological conditions affecting muscle tissue, such as cachexia, aging, insulin resistance, heart failure, and muscular dystrophies [11,12].

How a single signaling molecule could be involved in diverse and opposite processes related to the metabolic response to exercise, inflammation, muscle regeneration, and cancer-associated cachexia remained elusive for a long time. One important consideration related to this issue is the length of time the circulating IL-6 is actually elevated, a chronic response versus a somewhat brief increase that returns to baseline [13]. Supporting this thesis, we previously demonstrated that increased levels of IL-6 exacerbated muscle wasting in dystrophic mice, more closely approximating human pathology [14,15,16,17]. We further highlighted a correlation between the systemic deregulation of IL-6 abundance and the local impairment of the physiologic muscle redox balance, indicating that skeletal muscle tissue is sensible to dysregulated levels of IL-6 cytokine, which induce maladaptive responses [18,19]. A wealth of other studies demonstrated that chronic administration of IL-6 to otherwise healthy mice or rats through systemic administration, in vivo electroporation, or infused directly into skeletal muscle, can decrease muscle mass, muscle protein content, or slow the rate of muscle growth [13,20,21]. However, these effects appear related to the level of circulating IL-6, the type of administration, and the age of the rodent. Thus, whether IL-6 signaling has a role in the induction of muscle atrophy and wasting remains controversial, along with to be determined if supraphysiological levels of IL-6 regulate wasting mechanisms similar to the low-level chronic inflammation seen in many cachectic conditions as well as during aging [13,22,23].

In this context, the purpose of this study was to analyze and dissect the potential impact of IL-6 cytokine on muscle growth and maintenance. To this purpose, we used the NSE/IL-6 transgenic mouse model that associates a growth defective phenotype with high circulating levels of IL-6 since birth [21]. In this model, we comparatively investigated the effects of high levels of IL-6 cytokine on muscle homeostasis and in the induction of altered phenotype. Our study demonstrated that skeletal muscle groups display a different vulnerability in response to the chronically elevated levels of IL-6 in the bloodstream, with fast-twitch glycolytic fibers that resulted more vulnerable than slow-twitch oxidative fibers. The chronic increased plasma levels of IL-6 affect skeletal muscle homeostasis, promotes alteration in skeletal muscle growth, functional performance, and metabolism.

## 2. Materials and Methods

### 2.1. Animal Models

Wild type C57Bl/6J mice and NSE/IL-6 transgenic mice of 1.5, 3.5 and 6 months of age and 6-month-old MCK/PGC-1α MCK/PGC-1α:IL-6 mice were used to perform the experiments included in the study, in accordance with the Italian Law on the Protection of Animals. The NSE/IL-6 murine model is a transgenic mouse overexpressing circulating IL-6 since early after birth [21]. MCK/PGC-1α (reported as PGC-1α) mice express the PGC-1α complementary DNA downstream of MCK promoter sequence [24]. MCK/PGC-1α:IL-6 (re-ported as PGC1α:IL-6) were generated by crossing NSE/IL-6 male mice with PGC-1α female mice. Animals were housed in ventilated cages at controlled temperature and humidity conditions, in accordance with the guidelines of the institutional animal facility.

### 2.2. Histological, Morphometrical and Immunofluorescence Analysis

Soleus muscles were isolated from 1.5, 3.5 and 6-month-old wild type and NSE/IL-6 mice, embedded in tissue freezing medium and snap frozen in nitrogen-cooled isopentane to obtain cryostat transversal sections. For general morphology and morphometry, hematoxylin and eosin staining was performed according to standard protocols. Images of cross-sections were captured with a digital camera (model Axioskop 2 plus; Carl Zeiss Microimaging, Inc., Jena, Germany), and processed using Axiovision 3.1 software. The cross-sectional area (CSA) of muscles and single myofibers were analyzed using ImageJ software (v.1.51j8; National Institutes of Health, Bethesda, MD, USA). 

For immunofluorescence analysis, 12 µm-thick muscle cryosections of 6-month-old wild type, NSE/IL-6, MCK/PGC-1α, and MCK/PGC-1α:IL-6 mice were immunostained using anti-laminin (L9393) antibody, monoclonal anti-myosin slow (M8421), monoclonal anti-myosin fast (M1570) from Sigma-Aldrich; Merck KGaA, Darmstadt, Germany and appropriate secondary fluorescent antibodies (Invitrogen; Thermo Fisher Scientific, Inc., Waltham, MA, USA). Photomicrographed sections were analyzed by ImageJ software to quantify the percentage of Type I and Type II fibers. 

### 2.3. RNA Extraction and Real Time PCR Analysis

Soleus muscles were isolated from wild type, NSE/IL-6, MCK/PGC-1α, and MCK/PGC-1α:IL-6 mice of 6 months of age. For RNA extraction, muscles were frozen in liquid nitrogen, powdered and homogenized in TRI Reagent (Sigma-Aldrich) by using Tissue Lyser (Quiagen, Hilden, Germany). The reverse transcription was performed by using QuantiTect Reverse Transcription Kit (Quiagen) according to the manufacturer’s instructions. Real time PCR analysis was performed on ABI PRISM 7500 SDS (Applied Biosystems; Thermo Fisher Scientific, Inc., Waltham, MA, USA) by using specific TaqMan assays (Atrogin1, MURF-1, PGC-1α, MEF2c, and AchR-γ; Applied Biosystems). Relative quantification was performed using and β-actin (Applied Biosystems, USA) as endogenous control. Data were analyzed using the 2-DDCt method and reported as mean fold change in gene expression relative to wild type.

### 2.4. Confocal Microscopy and Neuromuscular Junction Evaluation

Neuromuscular junction evaluation was performed in Tibialis anterior muscles derived from 6-month-old wild type and NSE/IL-6 and cryosectioned into 50-μm-thick slices with longitudinal orientation with respect to the central axis of the muscle. AchRs were labeled with Alexa Fluor 488–conjugated α-bungarotoxin (10 nM; Molecular Probes, Eugene, OR, USA). Z-stack images were obtained at sequential focal planes 2 μm apart using a confocal microscope (Laser Scanning TCS SP2; Leica Microsystems, Wetzlar, Germany). The images were acquired utilizing the Leica confocal software. Laser line was at 488 nm for bungarotoxin excitation. The images were scanned under a 20× objective. NMJs were analyzed in terms of fragmentation (Number fragments/NMJ) and topology of branching pattern (Number of branches/NMJ). Representative images are flattened projections of Z-stack series. Approximately 200 optical sections recovered in at least three different experiments performed in both samples were analyzed.

### 2.5. Functional Analysis

For ex vivo functional analysis, freshly isolated soleus muscles were dissected from 3.5 and 6-month-old NSE/IL-6 PGC-1α, PGC-1α:IL-6 and wild type mice. Each muscle was mounted in a temperature-controlled chamber containing a Krebs-Ringer bicarbonate buffer continuously gassed with a mixture of 95% O2 and 5% CO2. Tetanic forces have been measured as described in [25], and specific force was obtained by dividing the maximum force by the muscle Cross Sectional Area [26].

### 2.6. Statistical Analysis

Data were expressed as mean ± SEM and differences among experimental groups were assessed with Mann–Whitney test or Student’s *t* test, assuming two-tailed distributions. Differences among experimental groups and time points were assessed by two-way ANOVA. Data analysis was performed using GraphPad Prism Software (San Diego, CA, USA) and values of *p* < 0.05 were considered statistically significant. The sample size was predetermined based on the variability observed in preliminary and similar experiments and to ensure adequate power.

## 3. Results

### 3.1. Systemic Levels of IL-6 Induce Differential Impact on Distinct Muscular Districts

To investigate the effects of increased circulating levels of IL-6 on muscle growth and homeostasis, we monitored, at morphological and functional levels, the features of both wild type and NSE/IL-6 transgenic mice at different ages of postnatal life. At first, we evaluated the whole-body size of NSE/IL-6 and wild type mice at 1.5, 3.5 and 6 months of age and we observed a strong reduction of body mass in IL-6 mice compared with age- and sex-matched wild type littermates (Figure 1a). This supports the evidence that demonstrated a general growth defect in NSE/IL-6 mice compared to wild type littermates, leading to an adult size of IL-6 transgenic animals which is about 30–50% smaller than that of non-transgenic littermates [21]. 

Thus, we evaluated the growth rate of the different hindlimb muscles of both wild type and NSE/IL-6 mice at 1.5, 3.5, and 6 months of age, charting distinctive growth curves of muscle districts. We revealed that muscle groups from NSE/IL-6 mice displayed a decrease in growth rate comparable to the reduction in body weight (data not shown). Nevertheless, whether and how chronic circulating levels of IL-6 impinge the maintenance of differentiated phenotype of different muscle groups remained to be defined. In this context, we analyzed the absolute weight of NSE/IL-6 muscle groups with the aim to define the specific physio-pathological role exerted by circulating levels of IL-6 in the maintenance of differentiated muscle phenotype, once the growth process ended.

Interestingly, gastrocnemius, quadriceps, and extensor digitorum longus (EDL) muscle of NSE/IL-6 mice displayed a significant reduction in weight already at 1.5 months of age compared to wild type littermates (Figure 1c–e). In contrast, soleus muscle of NSE/IL-6 transgenic mice showed no changes at 1.5 month of age (Figure 1b), while displayed a severe reduction in muscle mass at 6 months of age (Figure 1b). These data indicate that increased levels of circulating IL-6 might exert a differential impact on tissue homeostasis in several muscle types, early impinging the gain of muscle mass during the postnatal growth and subsequently affecting the maintenance of muscle mass during adulthood.

To support this evidence and to avoid the confounding action of chronic accumulation of IL-6 on postnatal growth, we focused our analysis on soleus muscle, in which the increased circulating levels of IL-6 did not significantly alter the early phase of muscle growth but induces a reduction in muscle mass in the adulthood. 

### 3.2. IL-6 Overexpression Impinges Muscle Growth at Early Stage of Postnatal Life

The muscle growth rate has been assessed by analyzing the well-established morphometric parameters, such as the total muscle cross-sectional area (muscle CSA), the mean fiber cross-sectional area (fiber CSA) and the total number of fibers. It has been reported that the implementation of muscle mass occurring in the post-natal life can be initially associated with satellite cell proliferation and fusion to growing fibers and then to the enhancement of protein synthesis and sarcomeric adjunction [26,27]. Indeed, at the end of the intensive growth period, the postnatal muscle growth is characterized by an inverse correlation between muscle fiber number and muscle fiber size [28,29,30]. Morphological and histo-morphometric analysis (Figure 2a–e) revealed that NSE/IL-6 soleus muscle grows with a different rate compared to wild type muscle. In fact, between 1.5 and 3.5 months of age, in both wild type (WT) and NSE/IL-6 soleus muscle, we observed a significant increase of total cross-sectional area (29% in WT vs. 19% in NSE/IL-6) (Figure 2c); a different increase of myofiber cross-sectional area (30% in WT vs. 17% in NSE/IL-6) (Figure 2d) and a significant reduction in total number of fibers (20% in WT vs. 19% in NSE/IL-6) (Figure 2e). These data revealed that soleus muscles of 1.5- and 3.5-month-old NSE/IL-6 transgenic mice did not show significant difference in terms of total muscle CSA and single fibers CSA, compared to muscles of wild type mice, and thus it grows regularly, although with a different rate compared to wild type soleus muscle. 

### 3.3. IL-6 Overexpression Induces a Reduction in Muscle Mass in the Adulthood

In contrast, a significant difference between wild type and NSE/IL-6 transgenic muscle was observed at 6 months of age (Figure 3). Hematoxylin and eosin staining (Figure 3a) and morphometric analysis (Figure 3b–d) revealed a severe alteration of overall morphological parameters analyzed in the soleus muscle of 6-month-old NSE/IL-6 mice compared to wild type littermates. In particular, the reduction in muscle mass of NSE/IL-6 mice (Figure 1b) was associated with a significant reduction in muscle CSA (Figure 3b), in single myofibers’ CSA (Figure 3c), and in the total number of fibers (Figure 3d). To substantiate these data, we further analyzed the frequency distribution of myofiber CSA, in both wild type and NSE/IL-6 soleus muscles. In accordance with the histological analysis, this morphometric parameter was not different between wild type and transgenic mice at 1.5 months of age (Figure 3e). In contrast, at 3.5 (Figure 3f) and, most strikingly, at 6 months of age (Figure 3g) the frequency distribution of myofibers revealed a significant shift of the median value towards smaller rates of fiber size in NSE/IL-6 mice compared to wild type littermates. 

Altogether these data suggest that increased levels of IL-6 induce a marked reduction of the muscle growth rate at early stage of postnatal life and significantly impair muscle homeostasis and promote a severe muscle atrophy in adulthood.

To substantiate this hypothesis, we analyzed skeletal muscle performance in NSE/IL-6 transgenic mice compared to wild type littermates. In particular, we performed a direct comparison of mechanical parameters of soleus muscle in both wild type and transgenic mice at 3.5 and 6 months of age. The data revealed a significant decrease of tetanic force in 3.5-month-old NSE/IL-6 mice compared to wild type littermates (−43%, *p* < 0.05) (Figure 3h), whereas no differences were noted in specific force (Figure 3i). In contrast, at 6 months of age, muscle atrophy of transgenic mice was associated with a significant reduction in both tetanic and specific force (Figure 3j,k), indicating that increased circulating levels of IL-6 induce muscle atrophy and wasting and severely affects the functional parameters of adult skeletal muscle. 

### 3.4. IL-6 Overexpression Alters Skeletal Muscle Fibers Composition 

In line of this evidence, we analyzed relevant molecular markers of muscle atrophy [31], namely the muscle-specific E3 ubiquitin ligases Muscle RING Finger-1 (MURF-1) and MAFbx/atrogin-1. Real time PCR analysis revealed a significant increase in the expression levels of both MuRF1 and Atrogin-1 in 6-month-old NSE/IL-6 soleus muscle compared to the muscle of wild type littermates (Figure 4a,b).

Moreover, we aim to define the potential mechanism by which soleus muscle, which muscle mass and function remains stable during early life, suddenly succumb in adulthood. Of note, it has been demonstrated that slow-twitch oxidative fibers are more resistant to damage and to a variety of atrophic conditions than fast-twitch glycolytic fibers [32]. We hypothesized that increased circulating levels of IL-6 induce a shift in the typology of soleus muscle fibers from slow to more vulnerable fast-twitch fiber.

To support this hypothesis, we performed immunofluorescence analysis for the isoforms of slow myosin heavy chain (slow-MyHC) and fast myosin heavy chain (fast-MyHC) in the soleus muscle of 6-month-old NSE/IL-6 and wild type mice (Figure 4c). The quantitative analysis of fiber composition revealed a significant increase in fast type II fibers with a concomitant decrease in the slow type I fibers in the soleus muscle of NSE/IL-6 transgenic mice compared to the soleus muscle of wild type littermates (Figure 4c, right panels). In addition, we analyzed, by quantitative RT-PCR, the expression of peroxisome proliferator-activated receptor γ coactivator 1 PGC-1α, a key regulatory factor of fiber type I determination [24,33] and a regulator of fiber-type switching from glycolytic towards more oxidative fibers [34] and its coactivator MEF2c, necessary to drive transcription of slow fiber gene expression [24,33]. Figure 4d,e showed that PGC-1α and MEF2c expression were strongly down-regulated in the soleus muscle of 6-month-old NSE/IL-6 transgenic mice compared to wild type littermates. These results suggest that IL-6 overexpression promotes a metabolic shift of soleus muscle fibers towards a less oxidative phenotype, rendering the muscle more susceptible to damage. 

To strengthen our data about the involvement of a slow-to-fast shift in muscle composition as a possible mechanism underlaying the decline of muscle mass and function observed in NSE/IL-6 mice, we enhanced PGC-1α expression in skeletal muscle of the IL-6 transgenic model. In particular, the NSE/IL-6 transgenic mouse model was bred with MCK/PGC-1α mice carrying a transgene expressing PGC-1α under the control of the muscle creatine kinase promoter [24]. Since PGC-1α activity triggers the conversion of muscle fibers from a fast to a slow phenotype [24], we analyzed whether overexpression of PGC-1α in IL-6 transgenic mice could prevents the shift in fiber type composition. Immunofluorescence analysis revealed that forced expression of PGC-1α enhanced the percentage of slow-twitch oxidative fibers in the soleus muscle of 6-month-old NSE/IL-6 mice (Figure 4f). Thus, to determine whether the restored muscle fiber composition induced by PGC-1α induction in NSE/IL-6 muscle can also improve force-generation capacity, a direct comparison of mechanical parameters was performed for the soleus muscle of 6-month-old wild type, NSE/IL-6, PGC-1α, and PGC-1α:IL-6 transgenic mice. PGC-1α expression was associated with a significant increase in tetanic and specific force generation, (Figure 4g,h), suggesting that forced expression of PGC-1α counteracts the negative effects exerted by increased plasma level of IL-6 on skeletal muscle. 

### 3.5. Increased Plasma Levels of IL-6 Induce Alterations in Neuromuscular Junction Morphology and Stability

The alteration in the heterogeneity of NSE/IL-6 muscle fibers would indicate an alteration in motor neuron activity, since motor neurons are known to regulate the properties of the myofibers they innervate by selective activation of fiber specific gene expression. Based on the reciprocal influence between muscle and nerve, related changes in skeletal muscle structure and function have been associated with pathologic changes in the morpho-physiology of neuromuscular junctions (NMJ) [35,36].

To evaluate whether the changes in muscle fiber type, that characterized IL-6 mice, are associated with signs of denervation and alteration of neuromuscular junction (NMJ), we analyzed by confocal microscopy, NMJ morphology of soleus muscle at 6 months of age. High resolution confocal microscopy analysis revealed that in adult wild type muscle, as expected, the AchR clusters were organized in a classical pretzel-shaped structure (Figure 5a). In contrast, in NSE/IL-6 soleus muscle the postsynaptic endplate/compartment appeared less ramified and, sometimes, fragmented (Figure 5a). Quantitative analysis corroborated the histological observations and revealed a significant deterioration of NMJ morphology in adult NSE/IL-6 mice compared to wild type littermates (Figure 5a, right panels).

To support the hypothesis of an NMJ destabilization, we analyzed the expression of the fetal gamma-subunit of the acetylcholine receptor (AchR-γ). Indeed, the re-expression of the fetal subunit during adulthood has been extensively described in denervated mice [37,38,39], were a strictly correlation with increasing gamma subunit mRNA levels has been reported [37,40]. As shown in Figure 4b the expression levels of AchR-γ subunit were significantly increased in the muscle of 6-month-old NSE/IL-6 mice compared to wild type littermates. Besides its recognized metabolic role, PGC-1α seems to be also involved in the maintenance of the NMJ, by exerting a neuroprotective effect [36,41]. Of note, forced expression of PGC-1α, reduced the expression of the AchR-γ subunit in the muscle of PGC-1α:IL-6 transgenic mice (Figure 5b).

## 4. Discussion

Interleukin 6 (IL-6) is a pleiotropic cytokine that is produced by different cell types inducing different intracellular signaling pathways. Being recognized as a myokine, IL-6 is locally and transiently produced in response to exercise and injury, and it plays an important role in satellite cell proliferation and muscle growth [10]. Indeed, IL-6 signaling has been associated with the stimulation of hypertrophic muscle growth and myogenesis through regulation of the proliferative capacity of muscle stem cells [3,42]. Nevertheless, circulating IL-6 levels are normally very low or undetectable, and are markedly and persistently increased in several diseases associated with inflammation, inducing the transition from an acute to a chronic inflammatory response [43]. Elevated levels of IL-6 have been detected in the blood of patients with microbial infections, autoimmune diseases, neoplasia, and muscular dystrophy [19,44,45,46,47,48,49]. Interestingly, epidemiological studies indicate that age-related decline of muscle mass and strength can be associated with increased plasma levels of IL-6, coupled to decreased levels of growth factors (IGF-1) [50,51]. However, although serum IL-6 heightening has been proposed as a risk factor for disability and mortality in several pathologic conditions, a direct correlation between selective increase of circulating IL-6 levels and muscle alterations has not been yet established because of contrasting results of different studies. 

In our study, to better define the impact of IL-6 on muscle homeostasis and growth, we took the advantage of the NSE/IL-6 transgenic mouse model that associates a growth defective phenotype with high circulating levels of IL-6 since birth [21]. NSE/hIL-6 mice show a marked reduction of the general growth rate during postnatal life. The reduction in growth rate leads to an adult size, which is about 30–50% smaller than that of non-transgenic littermates. In our study, we aimed also to evaluate the growth rate of the different hindlimb muscles of both wild type and NSE/IL-6 mice at 1.5, 3.5, and 6 months of age, charting distinctive growth curves of muscle groups, namely gastrocnemius, EDL, quadriceps, and soleus, based also on the approach that measure the postnatal muscle growth, which is characterized by an inverse correlation between muscle fiber number and muscle fiber size [29]. Our study reveals that all muscle groups from NSE/IL-6 mice displayed, although with a different rate, a decrease in growth rate comparable to the reduction of the body weight.

Nevertheless, whether and how chronic circulating levels of IL-6 also impinge the maintenance of differentiated muscle phenotype remained to be defined.

Interestingly, a significant difference between wild type and NSE/IL-6 transgenic muscle was observed in the soleus muscle of NSE/IL-6 mice at 6 months of age, suggesting that, beside the early defect in muscle growth rate, the adult soleus muscle displayed an atrophic phenotype in adulthood. In particular, the reduction in muscle mass of NSE/IL-6 mice was associated with a significant reduction in muscle CSA, in single myofibers’ CSA, and in the total number of fibers. 

Of note, the critical test to discriminate between defect in growth rate and muscle atrophy was obtained by functional test. The data revealed a significant decrease of tetanic force of soleus muscle of 3.5-month-old IL-6 mice, compared to wild type littermates, whereas no differences were noted in specific force. This suggests that the young soleus muscle, although displayed a reduced muscle growth rate, can function properly. In contrast, at 6 months of age muscle atrophy of transgenic mice was associated with a reduction in tetanic force, accompanied with a significant reduction in specific force, which is a measure of the ratio between force developed by the muscle and muscle mass. These data were supported by a significant increase in the two relevant molecular markers of muscle atrophy, namely the muscle-specific E3 ubiquitin ligases Muscle RING Finger-1 (MURF-1) and MAFbx/atrogin-1.

Soleus muscle has been described as slow muscle that, thanks to the oxidative metabolism and high levels of markers of mitochondrial biogenesis, is more protected against damage stimuli [52]. The recognition that skeletal muscles are composed of fiber types that differ in structure, molecular composition, and functional properties has contributed to our understanding of muscle physiology and plasticity [53]. Indeed, it is well known that muscle metabolism is closely associated with the sensibility or otherwise resistance to different stimuli. It has been reported that several muscle pathologies, including sarcopenia, cachexia, muscular dystrophies, and ALS, the fastest muscle phenotype is more severely compromised if compared with slow-twitch muscles. Moreover, a fiber-type shift in the direction IIb → IIx → IIa → I mitigate the progression of muscular dystrophy [54,55]. Interestingly, the significant muscle atrophy observed in the soleus muscle of IL-6 mice, was associated with a shift in fiber type composition, from slow-type I to fast-type II fibers, which might render the muscle more susceptible to damage. However, it has been extensively described that aging-related changes in muscle composition, with reference to the soleus muscle, involve the preferential loss of glycolytic fibers being slow fibers more resistant to atrophy and wasting [52,56,57]. The opposite shift observed in NSE/IL-6 mice can be justified by a dual action of deregulated levels of IL-6 on muscle fibers. It is well known that IL-6 can directly increase glucose metabolism in skeletal muscle and other cell types [58,59,60]. Moreover, we previously reported that elevated circulating levels of IL-6 can locally impair the redox balance of muscle tissue by enhancing pro-oxidant conditions and impinging the anti-oxidant response [18]. Based on these observations we can speculate that chronically elevated levels of IL-6 can induce muscle atrophy by promoting the shift through a more glycolytic metabolism in the soleus muscle of NSE/IL-6 mice, leading to an enhanced percentage of fast fibers, which are known to show greater atrophy than the type I oxidative fibers [61,62,63]. Furthermore, the redox alterations induced by high-level IL-6 can exacerbate wasting mechanisms by affecting both fast fibers, which present a lower antioxidant defense, and leading undefended slow-oxidative fibers, which are known to be subject to elevated levels of ROS because of their oxidative metabolism.

The greater sensitivity of the type II fibers may be due to their lower content of PGC-1α and others mitochondrial biogenesis markers, as suggested by the finding that muscle from limbs over-expressing PGC-1α was better able to maintain force production during a fatigue protocol when compared to control muscle, reflecting an improved endurance capacity [52,64]. Indeed, we found that soleus muscle of 6-month-old IL-6 mice has a strong reduction of both tetanic and specific force compared to wild type littermates, and a significant down-regulation of PGC-1α and its coactivator MEF2c, providing evidence that the reduction in the slow-mediated pathway might render the soleus muscle more sensitive to damage and fatigue. The changes in muscle fiber type and force observed in NSE/IL-6 transgenic muscle were also associated with signs of denervation and alteration of neuromuscular junction (NMJ), suggesting an alteration in the functional interplay between muscle and nerve, a typical feature of sarcopenia. Further associations of age-related signs of denervation have been established in animal models, including alterations in oxidative stress and a reduction in the number of key maintenance proteins such as PGC-1α [65]. Interestingly, the forced expression of PGC-1α into the muscle of NSE/IL-6 transgenic mice increased the percentage of slow-twitch oxidative fibers and induced an increased muscle performance and NMJ stability with respect to controls.

Taken together, the results of this study highlight a role for IL-6 in promoting the development of age-related alterations of muscle tissue. High-levels of IL-6 impinge early muscle growth during postnatal life and induce muscle atrophy and wasting in adulthood. This means that a long-lasting persistence of IL-6 into circulation can represent a risk, inducing a maladaptive mechanism that renders muscle tissue more susceptible to develop morpho-functional alterations.

## Figures and Tables

**Figure 1 cells-10-01816-f001:**
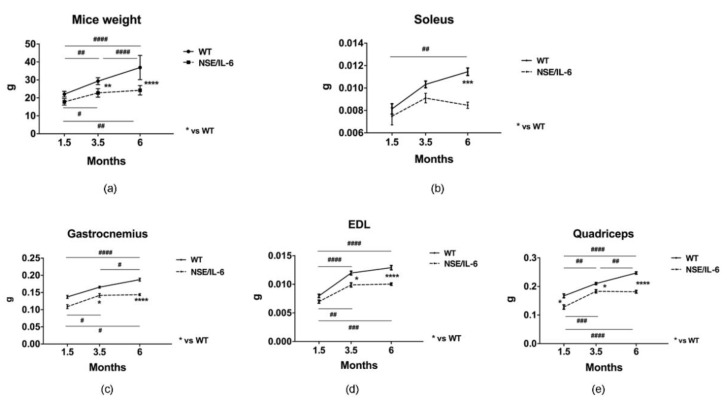
IL-6 overexpression affects muscle growth and size in distinct muscle groups. (**a**) Body weight growth curves. Wild type (WT) and NSE/IL-6 mice were measured to determine the change in body weight from 1.5 to 6 months of age. *n* = 9; ** *p* < 0.01, **** *p* < 0.0001, ^#^ *p* < 0.05, ^##^ *p* < 0.01, ^####^ *p* < 0.0001. (**b**–**e**) Growth curve of soleus (**b**), gastrocnemius (**c**), extensor digitorum longus (EDL) (**d**), and quadriceps muscles (**e**) of indicated genotype.. *n* = 12; * *p* < 0.05, *** *p* < 0.001; **** *p* < 0.0001, ^#^ *p* < 0.05, ^##^ *p* < 0.01, ^###^ *p* < 0.001, ^####^ *p* < 0.0001. All measurements are presented as means ± SEM. Statistical significance assessed by two-way ANOVA followed by multiple comparisons test.

**Figure 2 cells-10-01816-f002:**
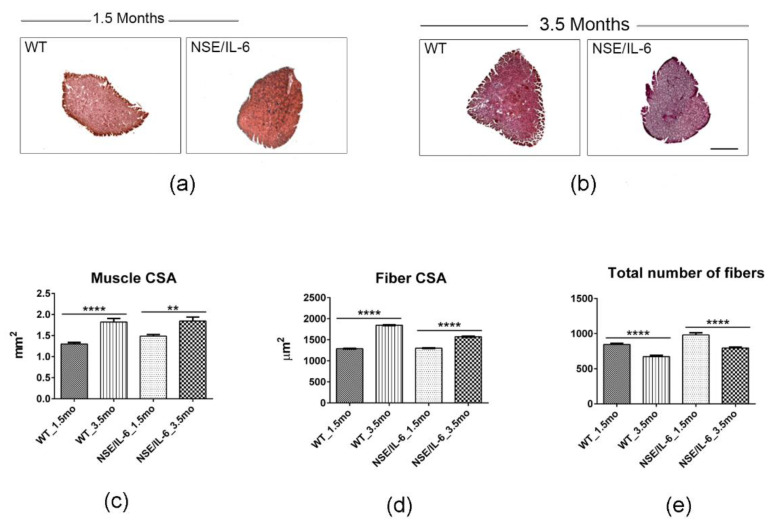
IL-6 overexpression significantly impairs muscle growth during the postnatal life. (**a**,**b**) Hematoxylin and eosin staining of transverse section of soleus muscles from indicated genotypes at 1.5 and 3.5. Scale bar 500μm. (**c**–**e**) The graphs show quantification of total cross-sectional area (muscle CSA) (**c**), fibers-cross sectional area (Fiber CSA) (**d**) and total number of fibers (**e**) in soleus muscles of wild type (WT) and NSE/IL-6 (IL-6) mice at 1.5 and 3.5 months of age. Data are represented as average ± SEM. *n* = 10 mice; ** *p* < 0.01, **** *p* < 0.0001. Statistical significance assessed by two-way ANOVA.

**Figure 3 cells-10-01816-f003:**
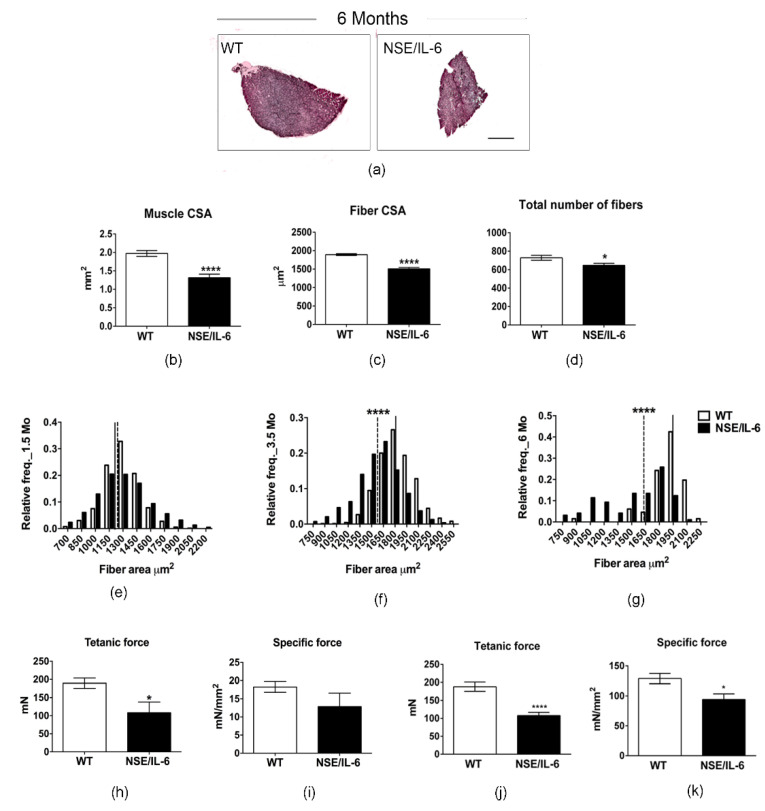
IL-6 overexpression promotes muscle atrophy and wasting and affects the functional performance of skeletal muscle in adulthood. (a) Hematoxylin and eosin staining of transverse section of soleus muscles from indicated genotypes at 6 months of age. Scale bar 500 μm.(**b**–**d**) Analysis of morphologic and morphometric parameters in transgenic (NSE/IL-6) and wild type (WT) soleus muscle at 6 months of age. Data are represented as average ± SEM. *n* = 10; * *p* < 0.05, **** *p* < 0.0001. Statistical significance assessed by unpaired Student’s *t*-test. (**e**–**g**) Frequency distribution of myofiber-cross sectional area in transgenic (NSE/IL-6) and wild type (WT) soleus at 1.5 (**e**), 3.5 (**f**), and 6 (**g**) months of age. Data are represented as medians. *n* = 10; **** *p* < 0.0001. Statistical significance assessed by Mann–Whitney Rank Sum Test. (**h**–**k**) Physiological properties of soleus muscle from 3.5 (**h**,**i**) and 6 (**j**,**k**)-month-old wild type (WT) and NSE/IL-6 mice. Tetanic force and Specific force. Data are represented as average ± SEM. *n* = 15; * *p* < 0.05, **** *p* < 0.0001. Statistical significance assessed by unpaired Student’s *t*-test.

**Figure 4 cells-10-01816-f004:**
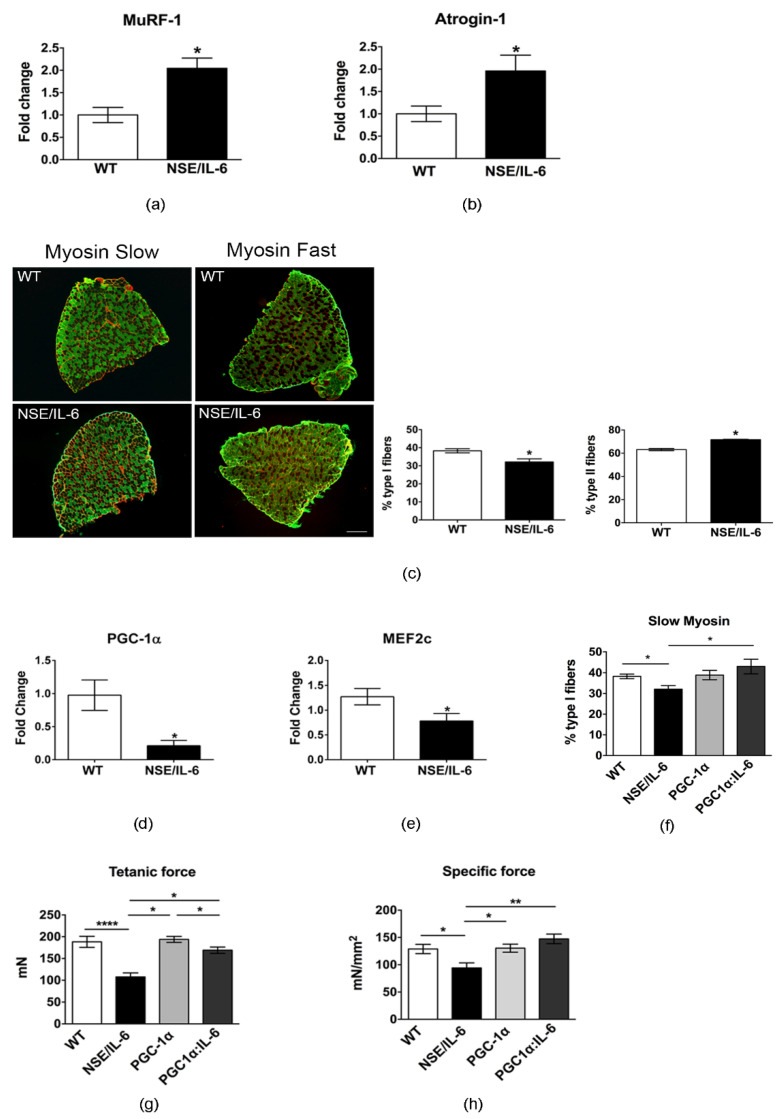
IL-6 overexpression impinges the heterogeneity of muscle fibers. (**a**,**b**) Real time PCR of ubiquitin-proteasome markers, Atrogin-1 and MuRF1 in soleus of wild type (WT) and NSE/IL-6 mice at 6 months of age. Values are expressed as fold change variations relative to WT and are presented as means ± SEM and *n* = 4; * *p* < 0.05. Statistical significance assessed by Mann–Whitney Rank Sum Test. (**c**) Immunofluorescence staining of wild type (WT) and NSE/IL-6 soleus muscles immunolabeled with either a mouse anti-myosin slow or a mouse anti-myosin fast antibody at 6 months of age. Scale bar 250μm. At the right, graphs showing the percentage of positive type I (left) or type II (right) muscle fibers. Data are represented as average ± SEM. *n* = 4 * *p*< 0.05. Statistical significance assessed by Mann–Whitney Rank Sum Test. (**d**,**e**) Real time PCR for the expression of PGC-1α and MEF2c in soleus of wild type (WT) and NSE/IL-6 mice at 6 months of age. Values are presented as means ± SEM and *n* = 4; * *p* < 0.05. Statistical significance assessed by Mann–Whitney Rank Sum Test. (**f**) Quantification of immunofluorescence staining of wild type (WT), NSE/IL-6, PGC-1α and IL-6:MCK/PGC-1α (PGC1α:IL-6) soleus muscles immunolabeled with a mouse anti-myosin slow antibody at 6 months of age. Data are represented as average ± SEM. *n* = 3, * *p*< 0.05. Statistical significance assessed by Mann–Whitney Rank Sum Test. (**g**,**h**) Physiological properties of soleus muscle from 6-month-old wild type (WT), NSE/IL-6, PGC-1α and IL-6:MCK/PGC-1α (PGC1α:IL-6) mice: Tetanic force and Specific force. Data are represented as average ± SEM. *n* = 8; * *p* < 0.05, ** *p* < 0.01, **** *p* < 0.0001. Statistical significance assessed by unpaired Student’s *t*-test.

**Figure 5 cells-10-01816-f005:**
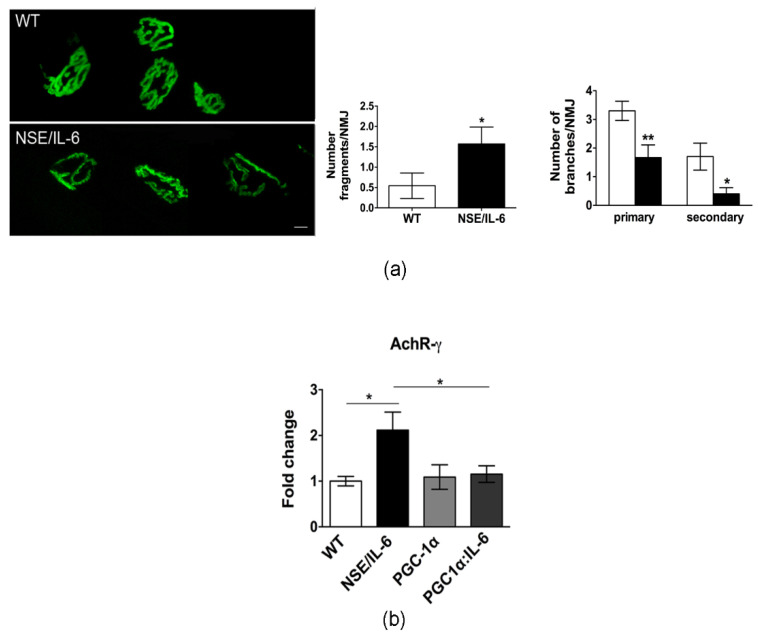
IL-6 overexpression affects neuromuscular junction (NMJ) destabilization. (**a**) Confocal microscopy of post-synaptic NMJ stained with a-bungarotoxin; representative images of one spatial series (from 0 to 50 µm) composed of 25 optical sections with a step size of 2 µm, from tibialis anterior of 6-month-old wild type and NSE/IL-6 mice. Scale bar 10 μm. Graphs showing the degree of NMJ fragmentation (Number fragments/NMJ) and the topology of branching pattern (Number of branches/NMJ). Data are represented as average ± SEM *n* = 3; * *p*< 0.05, ** *p* < 0.01. Statistical significance assessed by Mann–Whitney Rank Sum Test. (**b**) Real time PCR for the expression of AchR-γ in soleus wild type (WT) and NSE/IL-6, PGC-1α, and PGC1α:IL-6 mice at 6 months of age. Values are expressed as fold change variations relative to WT and are presented as means ± SEM and are *n* = 3; * *p* < 0.05. Statistical significance assessed by Mann–Whitney Rank Sum Test.

## Data Availability

The datasets used to support the findings of this study are included in the present article and are available from the corresponding author upon request.

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
