# Peer review of "Sustained Systemic Levels of IL-6 Impinge Early Muscle Growth and Induce Muscle Atrophy and Wasting in Adulthood"

_cells, 2021, doi:10.3390/cells10071816_

Round 1
Reviewer 1 Report
A brief summary
The authors report the long-term IL-6 effect on skeletal muscles using IL-6 overexpressing mice. The main findings are that: 1) muscle mass reduction in NSE/IL-6 mice accompanied with MuRF-1 and Atrogin-1 induction. 2) slow to fast fiber type shift in soleus of NSE/IL-6 mice. 3) impaired NMJ in NSE/IL-6 mice. In general, appropriate experimental approaches have been used. However, some sentences are difficult to understand. Revisions are suggested to render the manuscript acceptable.
Major comments
- In Figure 1, the atrophic phenotype of NSE/IL6 mice was described. Body weight and muscle weight of NSE/IL6 mice were decreased. Did the other tissues of NSE/IL-6 also decrease? Or is body length of NSE/IL6 mice different? It would be informative to include those data if authors have already obtained them.
- In Figure 2(e), soleus muscle weight is shown. Is the data of figure 2(e) the same data as Figure 1(b)? If the data is the same, authors should refer the Figure 1(b).
- In line275-278 of page8, it is written “a significant reduction in force derivative during single pulse stimulation (dF/dt) suggesting that IL-6 could modulate the fiber-type composition toward a faster phenotype (Figure 3a, bottom panel)”.
Fast type MyHCs have a higher ATPase activity, which enables fast contraction. So fast type muscles would have a higher force derivative compared with slow type muscles. The sentence above is not reasonable. If authors would like to insist on the logic, please include some references which support the concept “lower force derivative in faster muscle”.
Minor comments.
- In line219 of page7, it is written “Statistical 218 significance assessed by Mann-Whitney Rank Sum Test”. For statistical analysis of Figure 2(j), did authors use Mann-Whitney Rank Sum Test?
- In the bottom panel of Figure 3(a) and the right panel of Figure 3(e), the unit of Y-axis is “ms”. dF/dt would be “mN/ms”.
Reviewer 2 Report
This paper studied about the effect of high expression level of IL-6 on muscle formation. The authors analyzed the muscles from IL-6 overexpressed transgenic mice (NSE/IL-6 mice) and found that IL-6 induce muscle atrophy and then small muscle. This phenotype is partially rescued by crossing with PCG-1alpha that is related to energy consumption, overexpression mice. This is interesting, but there is something to consider about the size of NSE/IL-6 mice.
- All figure: the mark of figure (a, b, c,,,,) is better to be top left.
- Figure 1: The authors should consider about body length or bone length is essential. Because NSE/IL-6 mice is smaller than wild type, the muscle from NSE/IL-6 mice is obviously small size and light weight.
- Figure 2b: The authors should compare the data NSE/IL-6 mice vs wild type mice.
Reviewer 3 Report
Authors evaluated the roles of sustained elevated levels of IL-6 in muscle atrophy using animal models, and presented very interesting results. Manuscript was well-written and results partially supported well their hypothesis. However, several issues should be concerned to be published.
First, authors presented the several phenotypical and morphological data using soleus muscle (Type I slow-twitched muscle) to suggest the roles of IL-6 in aging-induced sarcopenia (gerokine). However, soleus muscle is 'anti-gravity muscle' that is not severely decreased by aging, rather white gastrocnemius (Type II fast-twitched muscle), for example, is a target of aging-induced sarcopenia. As authors presented, the results are not consistent to support this general phenotype during aging process of skeletal muscle. Authors should interpret their results more carefully, although authors tried to interpret and discussed their results briefly (page 11). In addition, it is not clear to interpret the age-dependent difference of muscle weight between animal groups according to muscle type, although authors argued that soleus muscle weight at early age (1.5 mo) was not different but others were different. Please re-analyzed the muscle weight data according to the time at analysis using correct statistical method. Authors used student t-test at each time point, however, two-way ANOVA (timexgroup) is more appropriate to compare groups with time-dependent changes, and then if significant, present group-comparison results.
Second, the effects of IL-6 over-expression on skeletal muscle biology is not clear. How authors argue that group difference is IL-6 -mediated skeletal muscle atrophy rather than IL-6 -mediated general growth retardation. Authors argued that IL-6 elevation causes skeletal muscle atrophy, however authors presented the muscle weight instead of muscle weight/body weight. How about the group comparison of muscle weight/body weight? Please re-analyze and discuss more about this concern.
Finally, authors presented figure including morphological data and graphs, but the size and resolution are not enough to be clearly reviewed by readers. Therefore, please replace the figures (H&E stain, IHC, IF, in particular) by high resolution pictures. Again, please re-analyze the data using appropriate statistical methods. Authors used unpaired t-test not non-parametric Mann-Whitney U test even the number of animals is too small (3-4). Please revise all p values using non-parametric methods, such as Mann-Whitney U test instead of unpaired t-test.
Minor points
Please indicate the number of animal per group clearly. What's the meaning of n≥3, or n≥6 ? Were authors use different animal numbers per each group? If in that case, please indicate the number of animal for each group in the graph, or clearly indicate in the text.
The graphs in each figure is too small, please present bigger graph with text large enough.
Authors should clearly describe the reason why the number of animals is different among results and graphs.
Round 2
Reviewer 2 Report
I think this paper is improved and suficient to be accpeted.
Reviewer 3 Report
Current revised version of manuscript was significantly improved and now is appropriate to be published.